# SEGMENT ANYTHING MODEL IS A GOOD TEACHER FOR LOCAL FEATURE LEARNING

## ABSTRACT

Local feature detection and description play an important role in many computer vision tasks, which are designed to detect and describe keypoints in "any scene" and "any downstream task". Data-driven local feature learning methods need to rely on pixel-level correspondence for training, which is challenging to acquire at scale, thus hindering further improvements in performance. In this paper, we propose SAMFeat to introduce SAM (segment anything model), a foundation model trained on 11 million images, as a teacher to guide local feature learning and thus inspire higher performance on limited datasets. To do so, first, we construct an auxiliary task of Pixel Semantic Relational Distillation (PSRD), which distillates feature relations with category-agnostic semantic information learned by the SAM encoder into a local feature learning network, to improve local feature description using semantic discrimination. Second, we develop a technique called Weakly Supervised Contrastive Learning Based on Semantic Grouping (WSC), which utilizes semantic groupings derived from SAM as weakly supervised signals, to optimize the metric space of local descriptors. Third, we design an Edge Attention Guidance (EAG) to further improve the accuracy of local feature detection and description by prompting the network to pay more attention to the edge region guided by SAM. SAMFeat's performance on various tasks such as image matching on HPatches, and long-term visual localization on Aachen Day-Night showcases its superiority over previous local features. The release code is available at supplementary material.

## 1 INTRODUCTION

Local feature detection and description is a basic task of computer vision, which is widely used in image matching (Balntas et al., 2017), structure from motion (SfM) (Schonberger & Frahm, 2016), simultaneous localization and mapping (SLAM) (Mur-Artal & Tardós, 2017), visual localization (Sattler et al., 2018a), and image retrieval (Wang et al., 2019) tasks. Traditional schemes such as SIFT (Lowe, 2004), and ORB (Rublee et al., 2011) based hand-crafted heuristics are not able to cope with drastic illumination and viewpoint changes (Balntas et al., 2017). Under the wave of deep learning, data-driven local feature learning methods (DeTone et al., 2018a; Tyszkiewicz et al., 2020) have recently achieved excellent performance. These methods require training local descriptors based on completely accurate and dense ground truth correspondences (Li & Snavely, 2018) between image pairs, but this type of data is difficult to collect. In addition, since local features are required to describe "any scenarios", it is impossible to cover all scenarios with a limited dataset. Recently, foundation models (Bommasani et al., 2021) have revolutionized the field of artificial intelligence. These models, trained on billion-size datasets, presented strong zero-shot generalization capabilities across a variety of downstream tasks. In this study, we advocate the integration of SAM Kirillov et al. (2023), a foundation model that is able to segment "anything" in "any scene", into the realm of local feature learning. This synergy enhances the robustness and enriches the supervised signals available for local feature learning, encompassing high-level category-agnostic semantics and detailed low-level edge structure information.

In recent years, several works have attempted to introduce pixel-level semantics of images (*i.e.* semantic segmentation) into local feature learning-based visual localization. Some methods utilized semantic information to filter keypoints (Xue et al., 2022) and optimize matching (Schönberger et al., 2018), while other works utilized semantic information (Xue et al., 2023) to guide the learning

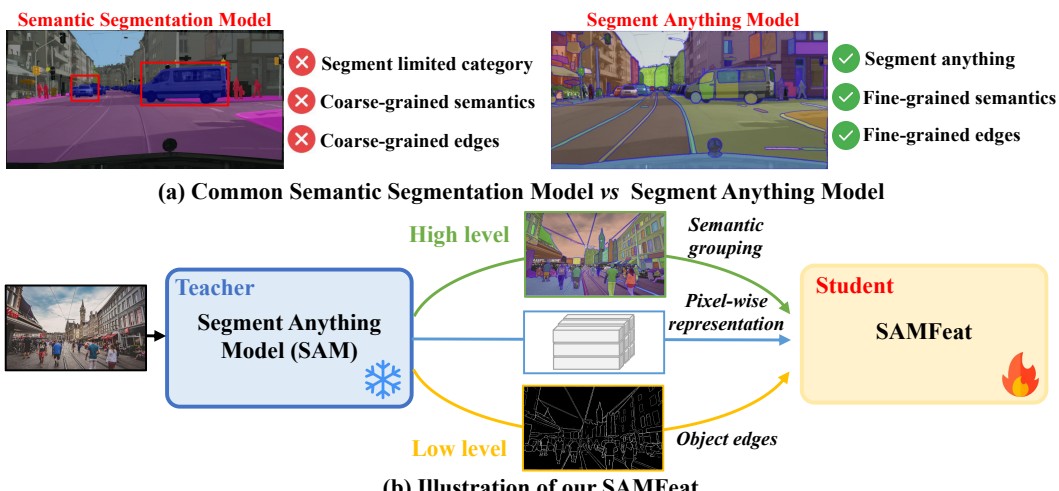

Figure 1: (a): Difference between segment anything model and common semantic segmentation model. (b): Schematic diagram of proposed SAMFeat.

of keypoints detection and improve the performance of the local descriptors in a specific visual localization setting by using feature-level distillation. However, these methods based on visual localization pipeline design are difficult to generalize to common feature matching tasks, as shown in Fig. 1 (a). On the one hand, semantic segmentation can only assign semantics to a few categories (*e.g.* cars, streets, people) which is difficult to generalize to generic scenarios. On the other hand, the semantic information for semantic segmentation is coarse-grained, *e.g.*, pixels of wheels and windows are given the same labels for a car. This is detrimental to mining the unique discriminative properties of local features.

The recent SAM (Kirillov et al., 2023) is a visual foundation model trained on 11 million images that can segment any objects based on prompt input. Compared to semantic segmentation models, SAM has three unique properties that can be used to fuel local feature learning. *i)* SAM is trained on a large amount of data, and therefore, can segment any object and can be adapted to any scene rather than being limited to street view. *ii)* SAM can obtain fine-grained part-level semantic segmentation results, thus allowing for more accurate modeling of semantic relationships between pixels. In addition, SAM can derive fine-grained category-agnostic semantic masks that can be used as semantic groupings of pixels to guide local feature learning. *iii)* SAM can detect more detailed edges, whereas edge regions tend to be more prone to keypoints and contain more distinguishing information. In our SAMFeat, we propose three special strategies to boost the performance of local feature learning based on these three properties of SAM. **First,** we construct an auxiliary task of Pixel Semantic Relational Distillation (PSRD) for distilling category-agnostic pixel semantic relations learned by the SAM encoder into a local feature learning network, thus using semantic discriminative to improve local feature description. **Second,** we develop a technique called Weakly Supervised Contrastive Learning Based on Semantic Grouping (WSC) to optimize the metric space of local descriptors using SAM-derived semantic groupings as weakly supervised signals. **Third,** we design an Edge Attention Guidance (EAG) to further improve the localization accuracy and description ability of local features by prompting the network to pay more attention to the edge region. Since the SAM model is only used as a teacher during training, our SAMFeat can efficiently extract local features during inference without burdening the computational consumption of the SAM encoder.

## 2 RELATED WORK

**Local Features and Beyond.** Early hand-crafted local features have been investigated for decades and are comprehensively evaluated in (Mikolajczyk & Schmid, 2005). In the wave of deep learning, many data-driven learnable local features have been proposed for improving detectors based on different focuses on (Mishkin et al., 2018; Barroso-Laguna et al., 2019), descriptors (Tian et al., 2017; Mishchuk et al., 2017a; Tian et al., 2019; Luo et al., 2019), and end-to-end detection and

description (Yi et al., 2016; Ono et al., 2018a; DeTone et al., 2018b; Revaud et al., 2019a; Dusmanu et al., 2019a; Revaud et al., 2019a; Luo et al., 2020a; Wang et al., 2022). Beyond localized features, some learnable advanced matchers have recently been developed to replace the traditional nearest neighbor matcher (NN) to get more accurate matching results. Sparse matchers such as SuperGlue (Sarlin et al., 2020) and LightGlue (Lindenberger et al., 2023) take off-the-shelf local features as input to predict matches using a GNN or Transformer, however, their time complexity scales quadratically with the number of keypoints. Dense matchers (Sun et al., 2021; Yu et al., 2023) compute the correspondence between pixels end-to-end based on the correlation volume, while they spend more memory and space consumption than sparse matchers (Xue et al., 2023). Our work centers on enhancing the efficiency and performance of an end-to-end generalized local feature learning approach. We aim to achieve performance comparable to advanced matchers while only using nearest-neighbor matching across various downstream tasks. This is particularly crucial in resource-constrained scenarios demanding high operational efficiency.

**Segment Anything Model.** Segment Anything Model (SAM) (Kirillov et al., 2023) has made significant progress in breaking the boundaries of segmentation, greatly promoting the development of foundation models for computer vision. SAM incorporates prompt learning techniques in the field of NLP to flexibly implement model building and builds an image engine through interactive annotations, which performs better in techniques such as instance analysis, edge detection, object proposal, and text-to-mask. SAM is specifically designed to address the challenge of segmenting a wide range of objects in complex visual scenes. Unlike traditional approaches that focus on segmenting specific object classes, SAM's primary objective is to segment anything, providing a versatile solution for diverse and challenging scenarios. Many works (He et al., 2023; Kristan et al., 2021) now build upon SAM for downstream vision tasks such as medical imaging, video, data annotation, *etc* (Zhang et al., 2023). Unlike them, we advocate for the application of SAM to local feature learning. To the best of our knowledge, our work is the first to apply SAM to segmentation-independent vision tasks. Since local feature learning has high operational efficiency requirements, it is not feasible to incorporate SAM directly into the pipeline, so we treat SAM as a teacher to bootstrap local feature learning, thus using SAM only in the training phase.

**Semantics in Local Feature Learning.** Prior to our work, semantics had only been introduced in the visual localization task to alleviate the limitations of low-level local features when dealing with severe image variations. Some early works incorporated semantic segmentation into the visual localization pipeline for filtering matching points (Huang et al., 2021; Hu et al., 2020), improving 2D-3D matching (Toft et al., 2018; Shi et al., 2020), and estimating camera position (Toft et al., 2017). Some recent works (Fan et al., 2022; Xue et al., 2023) have attempted to introduce semantics into local feature learning to improve the performance of visual localization. Based on the assumption that high-level semantics are insensitive to photometric and geometric, they enhance the robustness of local descriptors on semantic categories by distilling features or outputs from semantic segmentation networks. However, semantic segmentation tasks can only segment certain specific categories (*e.g.*, visual localization-related street scenes), preventing such approaches from generalizing to openworld scenarios and making them effective only on visual localization tasks. In contrast, we introduce SAM for segmenting any scene as a distillation object and propose the category-agnostic Pixel Semantic Relational Distillation (PSRD) scheme to enable local feature learning to enjoy semantic information in scenes beyond visual localization. In addition, we also propose Weakly Supervised Contrastive Learning Based on Semantic Grouping (WSC) and Edge Attention Guidance (EAG) to further motivate the performance of local features based on the special properties of SAM. Based on the above improvements, our SAMFeat makes it possible for local feature learning to more fully utilize semantic information and benefit in a wider range of scenarios.

## 3 METHODOLOGY

### 3.1 PRELIMINARY

**Segment Anything Model (SAM).** SAM (Kirillov et al., 2023) is a newly released visual foundation model for segmenting any objects and has strong zero-shoot generalization due to the fact that it is trained using 11 million images and 1.1 billion masks. Due to its scale, model distillation (Hinton et al., 2015) is deployed in this work. We freeze the weights of SAM and use its output as pseudoground truth to guide more accurate and robust local feature learning.

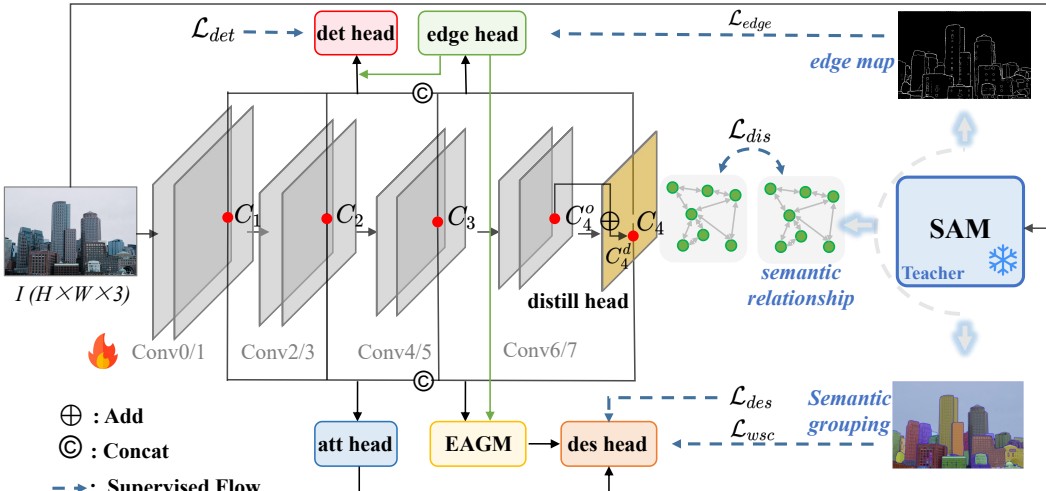

Figure 2: The overview of SAMFeat. Notice that SAM is only applied in the training phase, while there is no computational cost in the inference phase.

**Baseline.** To enhance clarity and simplicity, our SAMFeat takes the classic SuperPoint's (DeTone et al., 2018a) VGG-styled backbone and attention-weighted descriptor loss from its latest variant MTLDesc (Wang et al., 2022) as the baseline. The detailed network structure is shown in Figure 2. Specifically, we use a lightweight eight-layer VGG-style backbone network to extract feature maps. For a image $I$ with size $H \times W$, we concatenate multiscale feature map outputs $(C_1 \in \mathbb{R}^{H \times W \times 64}, C_2 \in \mathbb{R}^{\frac{1}{2}H \times \frac{1}{2}W \times 64}, C_3 \in \mathbb{R}^{\frac{1}{4}H \times \frac{1}{4}W \times 128}, C_4 \in \mathbb{R}^{\frac{1}{8}H \times \frac{1}{8}W \times 128})$ delivered to the keypoint detection head (*det head*), edge head (*edge head*), attention head (*att head*), and descriptor head (*des head*). In addition, we add a distillation head to distill the semantic representation of SAM to enhance the $C_4$ feature map. Each head consists of a lightweight $3 \times 3$ convolutional layer. We adopt SuperPoint's paradigm of using pseudo-labeled keypoints to train keypoint detection and using metric learning to optimize local descriptors. In particular, we adopt the attention-based descriptor optimization paradigm (*i.e.* $\mathcal{L}_{des}$ in Figure 2) proposed in the recent MTLDesc for local descriptor learning.

## 3.2 GIFTS FROM SAM

Shown in Figure 2, we input the image $I$ into the SAM (Kirillov et al., 2023) with frozen parameters and then simply processed to produce the following three outputs for guided local feature learning.

**Pixel-wise Representations Relationship**: SAM's image encoder trained from 11 million images is used to extract image representations for assigning semantic labels. The representation of the encoder outputs implies a valuable semantic correspondence, *i.e.*, pixels of the same semantic object are closer together. To eliminate the effect of specific semantic categories on generalizability, we adopt relations between representations as distillation targets. SAM's encoder outputs $\mathcal{F} \in \mathbb{R}^{\frac{1}{8}H \frac{1}{8}W \times C}$, where $C$ is the channel number for feature map. The pixel-wise representations relationship can be defined as $\mathcal{R} \in \mathbb{R}^{\frac{1}{8}H \frac{1}{8}W \times \frac{1}{8}H \frac{1}{8}W}$, where $\mathcal{R}(i,j) = \frac{\mathcal{F}(i) \cdot \mathcal{F}(j)}{|\mathcal{F}(i)||\mathcal{F}(j)|}$.

**Semantic Grouping**: We use the automatically generating masks function[1] of SAM to obtain fine-grained semantic groupings. Specifically, it works by sampling single-point input prompts in a grid over the image, and SAM can predict multiple masks from each of them. Then, masks are filtered for quality and deduplicated using non-maximal suppression (Kirillov et al., 2023). The semantic grouping of the output can be defined as $G \in \mathbb{R}^{H \times W \times N}$, where $N$ is the number of semantic groupings. Notice that semantic grouping differs from semantic segmentation in that each grouping does not correspond to a specific semantic category (*e.g.* buildings, car, and person).

**Edge Map**: The binary edge map $E \in \mathbb{R}^{H \times W \times 1}$ is derived directly [2] from the segmentation results of SAM, which highlights the fine-grained object boundaries.

---

[1]https://github.com/facebookresearch/segment-anything

[2]https://github.com/ymgw55/segment-anything-edge-detection

### 3.2.1 SAMFEAT

Thanks to the gifts of the foundation model, SAM, we are able to consider SAM as a knowledge-able teacher with intermediate products and outputs to guide the learning of local features. First, we employ Pixel Semantic Relational Distillation (PSRD) to distill the category-agnostic semantic relations in the SAM encoder into SAMFeat, thereby enhancing the expressive power of local features by introducing semantic distinctiveness. Second, we utilize the high-level semantic grouping of SAM outputs to construct Weakly Supervised Contrastive Learning Based on Semantic Grouping (WCS), which provides cheap and valuable supervision for local descriptor learning. Third, we design an Edge Attention Guidance (EAG) to utilize the low-level edge structure discovered by SAM to guide the network to pay more attention to these edge regions, which are more likely to be detected as keypoints and rich in discriminative information during local feature detection and description.

**Pixel Semantic Relational Distillation.** SAM aims to obtain the corresponding semantic masks based on the prompt, so the encoder output representation of SAM is rich in semantic discriminative information. Unlike semantic segmentation, SAM does not project pixels to a specified semantic category, so we resort to distilling the semantics contained in the encoder by exploiting the relative relationship between pixels (*i.e.*, pixel representations of the same object are closer together).

For SAM, we derive the relation matrix $\mathcal{R}$ using the features extracted by the fixed-parameter SAM encoder as described in Section 3.2. For our SAM-Feat, $C_4^o$ is exported from *Conv7* layer and then imported into the distillation head to get $C_4^d \in \mathbb{R}^{\frac{1}{8}H \times \frac{1}{8}W \times 256}$. Following the operations reported in Sec. 3.2, the semantic relation matrix of $C_4^d$ can be defined as $\mathcal{R}'$. See ap-

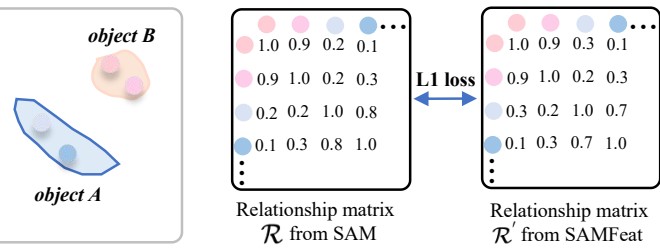

Figure 3: Schematic diagram of Pixel Semantic Relational Distillation.

pendix section 1.5 for more details. As shown in Figure 3, we distill the semantic relation matrix by imposing L1 loss in order to obtain semantic discriminativeness for $C_4^d$. $\mathcal{R}'$ and $\mathcal{R}$ are the corresponding student (SAMFeat) and teacher (SAM) relation matrix. Pixel semantic relational distillation loss $\mathcal{L}_{dis}$ can be defined as:

$$\mathcal{L}_{dis} = \frac{\sum_{i,j}^{(\frac{1}{8}H \times \frac{1}{8}W),(\frac{1}{8}H \times \frac{1}{8}W)} |\mathcal{R}_{i,j} - \mathcal{R}'_{i,j}|}{N}, \tag{1}$$

where $N$ is the number of matrix elements, *i.e.*, $(\frac{1}{8}H \times \frac{1}{8}W) \times (\frac{1}{8}H \times \frac{1}{8}W)$. Since PSRD is category-agnostic, it is possible to generalize local feature distillation semantic information to generic scenarios.

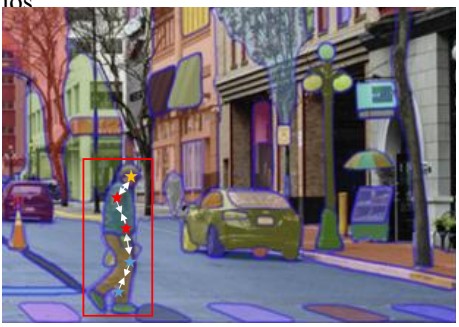

Figure 4: Example of Semantic Grouping. Different colored stars represent sampling points in different semantic groupings.

**Weakly Supervised Contrastive Learning Based on Semantic Grouping.** As shown in Figure 4, we use semantic groupings derived from SAM to construct weakly supervised contrastive learning to optimize the description space of local features. Our motivation is very intuitive: *i.e.*, pixels belonging to the same semantic grouping should be closer in the description space, and on the contrary pixels of different groupings should be kept at a distance in the description space. However, since two pixels belonging to the same grouping do not imply that their descriptors are the closest pair, forcing them to align will impair the discriminative properties of pixels within the same grouping. Therefore, semantic grouping can only provide weakly supervised constraints, and we maintain the discriminatory nature within the semantic grouping by setting a margin in optimization. Given the sampling points set $P \in \mathbb{R}^N$, the positive

sample average distance $D_{pos}$ can be defined as:

$$D_{pos} = \frac{1}{J} \sum_{i,j}^{J} \text{dis}(P_i, P_j), where\ G(i) = G(j)\ and\ i \neq j. \tag{2}$$

Here $\text{dis}(P_i, P_j)$ means calculate the Euclidean distance between the local descriptors corresponding to the two sampling points $P_i$ and $P_j$. $G(\cdot)$ denotes the indexed semantic grouping category. $J$ denotes the number of positive samples, noting that since $J$ is not consistent for each image, we take the average to denote the positive sample distance. Similarly, the negative sample average distance $D_{neg}$ can be defined as:

$$D_{neg} = \frac{1}{K} \sum_{i,j}^{K} \text{dis}(P_i, P_j), where\ G(i) \neq G(j), \tag{3}$$

where $K$ denotes the number of negative samples. Thus, the final $\mathcal{L}_{wsc}$ loss can be defined as:

$$\mathcal{L}_{wsc} = -\log(\frac{\exp(\max(D_{pos}, \text{M})/\text{T})}{\exp(\max(D_{pos}, \text{M}) + D_{neg})/\text{T})}), \tag{4}$$

where M is a margin parameter used to protect distinctiveness within semantic groupings, and T means the temperature coefficient.

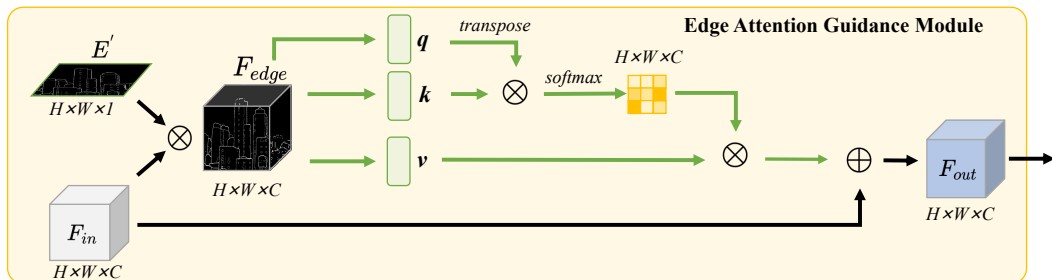

Figure 5: Details of Edge Attention Guidance Module.

**Edge Attention Guidance.** Edge regions are more worthy of the network's attention than mundane regions. On one hand, corner and edge points in the edge region are more likely to be detected as keypoints. On the other hand, the edge region contains rich information about the geometric structure thus contributing more to the discriminative nature of the local descriptor. To enable the network to better capture the details of edge areas and improve the robustness of descriptors, we propose the Edge Attention Guidance Module, which can guide the network to focus on edge regions. As shown in Figure 2, we first set up an edge head to predict the edge map $E'$ and use the SAM output of the edge map for supervision. The edge loss $\mathcal{L}_{edge}$ is denoted as

$$\mathcal{L}_{edge} = \sum_{i}^{H \times W} |E_i - E'_i|. \tag{5}$$

We then fuse the predicted edge map $E'$ into the local feature detection and description pipeline to bootstrap the network.

**1) Local Feature Detection**: As shown in Figure 2, we concat feature maps $\{C_1, C_2, C_3, C_4\}$ from different scales and then feed into the detection head to predict the heatmap for local feature detection. In particular, we enhance $C_3$ when performing the concat operation, *i.e.*, we pixel-wise dot product the edge map $E'$ into $C_3$, due to the fact that $C_3$ coincides with the shape of the edge map $E'$.

**2) Local Feature Description**: We filter the edge features by the predicted edge map and model the features of the edge region by a self-attention mechanism to encourage the network to capture the information of the edge region. Specifically, the predicted edge map $E'$ from the edge head, and the multi-scale feature maps $F_{in}$ extracted from the backbone are fed into the Edge Attention Guidance

Module. As shown in Figure 5, we first fuse $E'$ and $F_{in}$ by applying a pixel-wise dot product to obtain an edge-oriented feature map $F_{edge}$. Then we apply different convolutional transformations to the given $F_{edge}$ to get query $q$, key $k$, and value $v$ respectively. We then calculate the attention score using the dot product between query and key. Next, we use the *softmax* function on the attention score to obtain the attention weight, which is used to calculate the edge-enhanced feature maps with the value feature vector. Finally, the edge-enhanced feature maps and the $F_{in}$ are added to obtain the output feature maps $F_{out}$.

**Total Loss.** The total loss $\mathcal{L}$ can be defined as:

$$\mathcal{L} = \mathcal{L}_{det} + \mathcal{L}_{des} + \mathcal{L}_{dis} + \mathcal{L}_{edge} + \mathcal{L}_{wsc}. \tag{6}$$

$\mathcal{L}_{dis}$, $\mathcal{L}_{edge}$ and $\mathcal{L}_{wsc}$ are defined in section 3.2, while $\mathcal{L}_{det}$ is the cross entropy loss for supervised keypoint detection and $\mathcal{L}_{des}$ is the attention weighted triplet loss from MTLDesc (Wang et al., 2022) for optimizing the local descriptors. Individual weights for each loss are not assigned: each loss shares equal weights. This independence from hyper-parameters, again, shows the robustness of SAMFeat.

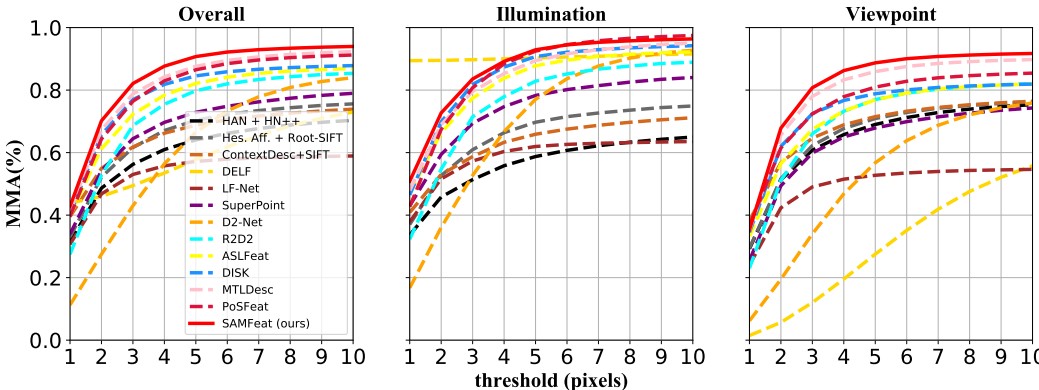

Figure 6: Comparisons on HPatches dataset with different thresholds Mean Matching Accuracy. Our SAMFeat achieves higher average local feature matching accuracy than other state-of-the-art at all thresholds.

## 4 EXPERIMENTS

**Implementation.** To generate our training data with dense pixel-wise correspondences, we rely on the MegaDepth dataset (Li & Snavely, 2018), a rich resource containing image pairs with known pose and depth information from 196 diverse scenes. Specifically, we use MTLDesc (Wang et al., 2022) [3] released megedepth image and the correspondence ground truth for training. In our experiment, we meticulously configured the parameters to establish a consistent and efficient training process. Hyper-parameters are set as follows. The learning rate of 0.001 enables gradual parameter updates, and the weight decay of 0.0001 helps control model complexity and mitigate overfitting. With a batch size of 14, our model processes 14 samples per iteration, striking a balance between computational efficiency and convergence. M and T are set to 0.07 and 5. Training spans 30 epochs to ensure comprehensive exposure to the data, with a total training time of 3.5 hours. By meticulously defining these parameters and configurations, we establish a robust experimental setup that ensures replicability and accurate evaluation of our model's performance. More detailed information about parameter tuning and ablation experiments can be found in the supplementary material.

**Image Matching.** We evaluate the performance of our method in the image-matching tasks on the most popular feature learning matching benchmark: HPatches (Balntas et al., 2017). The HPatches dataset consists of 116 sequences of image patches extracted from a diverse range of scenes and objects. Each image patch is associated with ground truth annotations, including key point locations, descriptors, and corresponding homographies. We follow the same evaluation protocol as in

---

[3]https://github.com/vignywang/MTLDesc

Table 1: Image Matching Performance Comparison on HPatches dataset.

| Methods | MMA @3 | AUC @5 |
|---|---|---|
| SIFT $_{IJCV2012}$ (Lindeberg, 2012) | 50.1 | 49.6 |
| HardNet $_{NeurIPS2017}$ (Mishchuk et al., 2017b) | 62.1 | 56.9 |
| DELF $_{ICCV2017}$ (Noh et al., 2017) | 50.7 | 49.7 |
| SuperPoint $_{CVPRW2018}$ (DeTone et al., 2018a) | 65.7 | 59.0 |
| Lf-net $_{NeurIPS2018}$ (Ono et al., 2018b) | 53.2 | 48.7 |
| ContextDesc $_{CVPR2019}$ (Luo et al., 2019) | 63.2 | 58.3 |
| D2Net $_{CVPR2019}$ (Dusmanu et al., 2019b) | 40.3 | 37.8 |
| R2D2 $_{NeurIPS2019}$ (Revaud et al., 2019b) | 72.1 | 64.1 |
| DISK $_{NeurIPS2020}$ (Tyszkiewicz et al., 2020) | 72.2 | 69.8 |
| ASLFeat $_{CVPR2020}$ (Luo et al., 2020b) | 72.2 | 66.9 |
| LLF $_{WACV2021}$ (Suwanwimolkul et al., 2021) | 74.0 | 66.8 |
| Key.Net $_{TPAMI2022}$ (Barroso-Laguna & Mikolajczyk, 2022) | 72.1 | 56.0 |
| ALIKE $_{TMM2022}$ (Zhao et al., 2022) | 70.5 | 69.0 |
| MTLDesc $_{AAAI2022}$ (Wang et al., 2022) | 78.7 | 71.4 |
| PoSFeat $_{CVPR2022}$ (Li et al., 2022) | 75.3 | 69.2 |
| SFD2 $_{CVPR2023}$ (Xue et al., 2023) | 70.6 | 64.8 |
| TPR $_{CVPR2023}$ (Wang et al., 2023) | 79.8 | 73.0 |
| **SAMFeat (Ours)** | **82.1** | **74.4** |

D2-Net (Dusmanu et al., 2019b), where eight unreliable scenes are excluded. To ensure an equitable comparison, we align the features extracted by each method through nearest-neighbor matching. A match is deemed accurate if its estimated reprojection error is lower than a predetermined matching threshold. The threshold is systematically varied from 1 to 10 pixels, and the mean matching accuracy (MMA) across all pairs is recorded, indicating the proportion of correct matches relative to potential matches. Subsequently, the area under the curve (AUC) is computed at 5px based on the MMA. The comparison between SAMFeat and other state-of-the-art methods on HPatches image matching is visualized in Figure 6. The MMA @3 threshold against other state-of-the-art methods under each threshold is listed in Table 1. SAMFeat achieved the highest MMA @3 even compared to the most updated feature learning model in 2023 top-tier conferences.

**Visual Localization.** To further validate the efficacy of our approach when dealing with intricate tasks, we assess its performance in the area of visual localization. This task involves estimating the camera's position within a scene using an image sequence and serves as an evaluation benchmark for local feature performance in long-term localization scenarios, without requiring a dedicated localization pipeline. We utilize the Aachen Day-Night v1.1 dataset (Sattler et al., 2018b) to showcase the impact on visual localization tasks. All methodologies are objectively compared on the official evaluation server to ensure fairness in the assessment. The assessment is carried out through The Visual Localization Benchmark, employing a predetermined visual localization framework rooted in COMLAP (Schonberger & Frahm, 2016). We tally the number of accurately localized images under three distinct error thresholds, namely (0.25m, 2°), (0.5m, 5°), and (5m, 10°), signifying the maximum allowable position error in both meters and degrees. We employ the Nearest Neighbors' matcher for a justifiable and equitable comparison among all methods. Referring to Table 2, we categorize current state-of-the-art methods into two categories: $G$ contains methods that are designed for general feature learning tasks; $L$ contains methods that are designed, tuned, and tested specifically for localization tasks, and they typically perform poorly outside of specific localization scenarios, as shown in Table 1. SAMFeat achieved the top performance among all general methods, while also revealing a competitive performance among methods that are designed specifically for visualization.

**Ablation Study.** Table 3 demonstrates the efficacy of the components within our network as we progressively incorporate Pixel Semantic Relational Distillation (PSRD), Weakly Supervised Contrastive Learning Based on Semantic Grouping (WCS), and Edge Attention Guidance (EAG). The

Table 2: Visual Localization Performance Comparison on Aachen V1.1. Category "L" means local feature methods specifically designed for visual localization tasks, and "G" means generalized local feature methods.

| Category | Method | Accuracy @ Thresholds (%) ↑ | |
|---|---|---|---|
| | | Day | Night |
| | | 0.25m,2°/0.5m,5°/5m,10° | |
| L | SeLF $_{TIP2022}$ (Fan et al., 2022) | – | 75.0 / 86.8 / 97.6 |
| | SFD2 $_{CVPR2023}$ (Xue et al., 2023) | 88.2 / 96.0 / 98.7 | 78.0 / 92.1 / 99.5 |
| G | SIFT $_{IJCV2012}$ (Lowe, 2004) | 72.2 / 78.4 / 81.7 | 19.4 / 23.0 / 27.2 |
| | SuperPoint $_{CVPRW2018}$ (DeTone et al., 2018b) | 87.9 / 93.6 / 96.8 | 70.2 / 84.8 / 93.7 |
| | D2-Net $_{CVPR2019}$ (Dusmanu et al., 2019a) | 84.1 / 91.0 / 95.5 | 63.4 / 83.8 / 92.1 |
| | R2D2 $_{NeurIPS2019}$ (Revaud et al., 2019a) | 88.8 / 95.3 / 97.8 | 72.3 / 88.5 / 94.2 |
| | ASLFeat $_{CVPR2020}$ (Luo et al., 2020a) | 88.0 / 95.4 / 98.2 | 70.7 / 84.3 / 94.2 |
| | CAPS $_{ECCV2020}$ (Wang et al., 2020) | 82.4 / 91.3 / 95.9 | 61.3 / 83.8 / 95.3 |
| | LISRD $_{ECCV2020}$ (Pautrat et al., 2020) | – | 73.3/ 86.9 / 97.9 |
| | DISK $_{NeurIPS2022}$ (Tyszkiewicz et al., 2020) | – | 73.8 / 86.2 / 97.4 |
| | PoSFeat $_{CVPR2022}$ (Li et al., 2022) | – | 73.8 / 87.4 / **98.4** |
| | MTLDesc $_{AAAI2022}$ (Wang et al., 2022) | – | 74.3 / 86.9 / 96.9 |
| | TR $_{CVPR2023}$ (Wang et al., 2023) | – | 74.3 / 89.0 / **98.4** |
| | SAMFeat (Ours) | **90.2 / 96.0 / 98.5** | **75.9 / 89.5** / 95.8 |

effectiveness of each component is reflected by the Mean Matching Accuracy at the pixel three threshold on the HPatches Image Matching task. Our baseline is trained using SuperPoint's (DeTone et al., 2018a) VGG-styled backbone along with its detector supervision and attention-weighted triplet loss (Wang et al., 2022) for descriptor learning. Following the addition of the PSRD, the model's performance notably improves due to better image feature learning. The introduction of the WCS further enhances accuracy by augmenting the discriminative power of descriptors with semantics. It demonstrates superior performance as it better preserves the inner diversity of objects by optimizing sample ranks. Lastly, the inclusion of the EAG bolsters the network's capability to embed object edge and boundary information, resulting in further enhancements in accuracy.

Table 3: Ablation Study on SAMFeat. ✓means denotes applied components.

| PSRD | EAG | WCS | MMA @3 |
|---|---|---|---|
| | | | 75.7 |
| ✓ | | | 78.6 |
| ✓ | ✓ | | 80.9 |
| ✓ | ✓ | ✓ | 82.1 |

## 5 CONCLUSION

In this study, We introduce SAMFeat, a local feature learning method that harnesses the power of the Segment Anything Model (SAM). SAMFeat encompasses three innovations. Firstly, we introduce Pixel Semantic Relational Distillation (PSRD), an auxiliary task aimed at distilling the category-agnostic semantic information acquired by the SAM encoder into the local feature learning network. Secondly, we present Weakly Supervised Contrastive Learning Based on Semantic Grouping (WSC), a technique that leverages the semantic groupings derived from SAM as weakly supervised signals to optimize the metric space of local descriptors. Furthermore, we engineer the Edge Attention Guidance (EAG) mechanism to elevate the accuracy of local feature detection and description. Our comprehensive evaluation of tasks such as image matching on HPatches and long-term visual localization on Aachen Day-Night consistently underscores the remarkable performance of SAMFeat, surpassing previous methods.

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
