# OpenReview forum: "Segment Anything Model is a Good Teacher for Local Feature Learning"
_ICLR.cc/2024/Conference — Submitted to ICLR 2024_

### Official Review · Reviewer_933k · 2023-10-23

**Soundness:** 3 good
**Presentation:** 3 good
**Contribution:** 3 good
**Rating:** 5
**Confidence:** 4

**Summary:**

This paper proposes a method called SAMFeat, which leverages a fundamental model called SAM (segment anything model) to improve local feature detection and description. The authors address the challenge of limited pixel-level correspondence data for training local feature learning methods.

To tackle this, they introduce an auxiliary task called Pixel Semantic Relational Distillation (PSRD). PSRD distills feature relations using category-agnostic semantic information learned by the SAM encoder, enhancing local feature description through improved semantic discrimination.

Additionally, the authors present Weakly Supervised Contrastive Learning Based on Semantic Grouping (WSC). This technique utilizes weakly supervised signals derived from SAM's semantic groupings to optimize the metric space of local descriptors, further enhancing the learning process.

To improve the accuracy of local feature detection and description, the authors propose an Edge Attention Guidance Module (EAGM). EAGM prompts the network to prioritize attention to the edge region, guided by the SAM model.

Experiments conducted on tasks such as image matching on HPatches and visual localization datasets like Aachen Day-Night demonstrate the superior performance of SAMFeat compared to previous local features.

**Strengths:**

+ This paper provides SAMFeat to leverage the SAM model to enhance local feature detection and description. The combination of distilling feature relations using category-agnostic semantic information, weakly supervised contrastive learning based on semantic grouping, and an edge attention guidance module is well-motivated.

+ The application of SAM as a teacher model to guide local feature learning is a creative combination that extends the existing understanding of feature learning approaches.

**Weaknesses:**

- In the section describing "Pixel Semantic Relational Distillation," further elaboration on the process of obtaining the relationship matrix from SAM would enhance the clarity and academic rigor of the paper. It is important to provide a more detailed explanation of the specific steps involved in acquiring the relationship matrix from SAM. This may include a description of the underlying mechanism used by SAM to capture and encode semantic information, followed by the extraction of relevant features for constructing the relationship matrix.

**Questions:**

- The authors should consider providing insights into the rationale behind choosing SAM as the source for extracting semantic information. Discussing the unique characteristics of SAM that make it suitable for distilling the semantic relationships within the given context would strengthen the paper's argument. For example, SEEM can also serve as a suitable baseline for addressing the task at hand.




------------ post rebuttal update ----------

Thanks for the feedback. After going through the rebuttal, I still think the generalization of the proposed method should be validated on other segment anything models such as SEEM. Therefore, I will keep the original rating.

---

> ### Author Response · Authors · 2023-11-19
> **Response to reviewer 933k**
>
> Dear Reviewer 933k,
>
> Thank you for your feedback. To further improve the soundness of our paper, we adopted your suggestions and polished writings in the main paper and Appendix. Still, we would like to address your concern here:
>
> **Weakness1:**
> Detailed explanation of the 'Pixel Semantic Relational Distillation' section.
>
> **A1:**
> Thanks to your suggestion, we have added detailed descriptions in the main paper and supporting materials.
>
> * We added a description of the process of extracting the relation matrix from SAM in Section "Pixel Semantic Relational Distillation," which we previously described in Section 3.2 "Pixel-wise Representations Relationship.
> * Due to limited space in the main paper, we have added a flowchart and detailed description of this process in the appendix.
> * In addition, we will release the code that precisely specifies the implementation details to facilitate further understanding by the reader.
>
> **SAM Encoder:**
> SAM uses an encoder (ViT-H [1]) to extract image features and then feeds them, along with the prompt, into a decoder to generate masks for the corresponding categories. Trained for pixel-wise segmentation tasks, the features extracted by the encoder are semantically discriminative, i.e., features extracted on the same semantic object pixel are more similar. Therefore, we advocate improving the distinctiveness of local feature matching by distilling semantic distinctiveness.
>
> **Semantic Relationship Matrix:**
> Unlike semantic segmentation, SAM does not project pixels to a specified semantic category, so the features extracted by the SAM encoder contain only category-independent semantic discriminative properties. This results in pixel features of the same class of objects (e.g., people) extracted from different images are not aligned. Direct distillation does not apply in this case (as reported in appendix section 1.4), so we resort to distilling the semantics contained in the encoder by exploiting the relative relationship between pixels (i.e., pixel representations of the same object are closer together). Specifically, SAM's encoder outputs after reshape $\mathcal{F} \in \mathbb{R}^{\frac{1}{8}H\frac{1}{8}W\times C}\$, where \(C\) is the channel number for the feature map. The pixel-wise representations relationship can be defined as $\mathcal{R} \in \mathbb{R}^{\frac{1}{8}H\frac{1}{8}W\times \frac{1}{8}H\frac{1}{8}W}$, where $\mathcal{R}(i,j)= \frac{\mathcal{F}(i) \cdot \mathcal{F}(j)}{|\mathcal{F}(i)||\mathcal{F}(j)|}$. Please check the appendix section 1.5 for more details.
>
>
>
> **Q2:**
> The rationale behind choosing SAM.
>
> **A2:**
> Thanks for your suggestion.
>
> **SAM's attractive properties:**
> As shown in the main paper Figure 1 (b), SAM has three good properties conducive to guided local feature learning.
>
> 1. Compared to traditional semantic segmentation, SAM is trained on a large amount of data and can **segment any object** without being restricted by the class of the training set. Both segmentation and local feature learning can be viewed as pixel-level representation learning, where segmentation is pixel-level classification while local features are used for matching. Thus we can improve the matching discriminativeness of local features by distilling the semantic discriminativeness of SAM pixel representations. However, traditional semantic segmentation models can only deal with the limited number of categories contained in the dataset and cannot be generalized to arbitrary scenarios.
>
> 2. **SAM yields semantic groupings at the part level**, and we can take advantage of this fine-grained semantic relationship to provide additional weakly supervised descriptor optimization supervision. In contrast, semantic segmentation tends to group all pixels of an object together, e.g., the pixels of the wheel and the window in Fig. 1 (a) are identified with the label of “car”, which is not conducive for use in guided learning of discriminative local features.
>
> 3. **SAM can detect more detailed edges**, whereas edge regions tend to be more prone to keypoints and contain more distinguishing information. We can achieve better local feature detection and description by making the network more focused on these edge regions.
>
> **Discussion:**
> Some of SAM's subsequent works have enhanced interactivity (SEEM), understanding (Semantic-SAM, LISA, SAM-CLIP), and efficiency (Fast SAM, Mobile SAM), but they do not additionally extend the three properties mentioned above that are relevant to our approach. Notice that our contribution is to improve local feature learning using a visual fundamental model without adding additional computational consumption. Our approach opens up the possibility of introducing visual fundamental models into the realm of geometric vision and inspires explorations using fundamental models other than SAM.
>
> [1] An image is worth 16x16 words: Transformers for image recognition at scale. ICLR 2021

---

> ### Public Comment · ~Will_Wu2 · 2023-11-30
>
> Thank the authors and the reviewer for conducting this interesting work and rebuttal discussion.
>
> The reviewer expressed a desire for validation on alternative models and stated that this consideration influenced his decision to maintain the original rating.
> In response to this, I would like to make the following comments based on my own interpretation and perspective on this work:
>
> (1) While the suggestion to explore SEEM as an alternative to SAM is intriguing, SAMFeat is explicitly designed based on the unique properties of SAM. SEEM faces two significant limitations: it has not been trained on large-scale data, and it lacks support for fine-grained part-level segmentation and edge detection. Therefore, replacing SAM with SEEM directly is not a feasible approach in the current context in my opinion.
>
> (2) I acknowledge the value and potential of extending the proposed method to other foundational models, as suggested by the reviewer. However, I believe that such exploration does not diminish the effectiveness or novelty of the authors’ approach. The key innovation lies in the clever and targeted application of SAM, and the exploration of other models does not alter the fact that SAM, and models like it, serve as effective teachers for the proposed method.
>
> I understand the importance of thorough evaluation and appreciate the reviewer's input. However, I kindly request a reconsideration of the impact of this specific comment on the overall rating. I believe that the proposed method's uniqueness and contributions remain valid within the context of SAM, and exploring alternative models can be a separate avenue for future research.

---

### Official Review · Reviewer_uqrh · 2023-10-28

**Soundness:** 3 good
**Presentation:** 2 fair
**Contribution:** 2 fair
**Rating:** 5
**Confidence:** 5

**Summary:**

This paper that proposes a method to utilize Segment Anything(SAM) Feature for local feature detection and description. Three key strategies are introduced:

1. Pixel Semantic Relational Distillation (PSRD): An auxiliary task that enhances local feature descriptions using category-agnostic semantic information from the SAM encoder.
2. Weakly Supervised Contrastive Learning Based on Semantic Grouping (WSC): A technique that employs semantic groupings from SAM as weak supervision to optimize the metric space of local descriptors.
3. Edge Attention Guidance (EAG): A design strategy that improves the accuracy of local feature detection by directing the network to focus more on edge regions, guided by SAM.

SAMFeat demonstrates good performance in image matching on HPatches, and and visual localization on Aachen V1.1 dataset, compared to previous local features.

**Strengths:**

1. This paper exlpored a way to release the power of Segment Anything Model (SAM) for distillation for local features. It shows the potential s of visual foundation models.
2. Experiment-wise, it reaches state of the art results for image matching for with different on HPatches dataset and visual localization task on Archen V1.1 dataset.
3. Authors provide open-source code

**Weaknesses:**

Although I believe in the soundness of the good results that the authors have demonstrated, a major issue that makes me skeptical is whether the contribution and novelty are substantial enough to warrant a full paper. Many of the techniques used in the paper are borrowed from other's implementation. For example, the Pixel Semantic Relational Distillation (PSRD) is to compare two similarity matrix which is a widely used knowledge distillation loss [1]. Then the semantic grouping is from the original SAM implementation. The contrastive loss is also very straightforward. The presentation of the paper lacks clarity in conveying the motivation behind technique used, making the paper seem more like a engineering solution for a specific task rather than a systematic method.

[1] F, Tung, et al. Similarity-Preserving Knowledge Distillation, ICCV19

**Questions:**

1. Continued from the weakness section, could you provide more intuition of each technique used in the paper? For example, the similarity-preserving style of knowledge distillation is used. Why can we use other distillation techniques?
2. Is SAM really a good teacher? Some works[1] have shown SAM has worse semantics than other pretrained vision model. Correct me if I am wrong, to my understanding, the proposed framework can be applied any pretrained model with good semantics like DINOv2[2], CLIP[3] and ect. Those framework has been proved to have very good features. What would be the results to distillate from those pretrained models for image matching and visual localization.
3. One minor question of the use of SAM. SAM can segment different levels of object due to ambiguity. For example, a person can be defined as an object or the cloth of this person can be defined as an object. When applying semantic grouping, how to define such grid propmt for SAM?
4. Can we directly use SAM feature for such image matching and visual localization task?
5. For the first line of table 3, how is it different with MTLDesc?

[1] Y,Liu, et al. Matcher: Segment Anything with One Shot Using All-Purpose Feature Matching, arXiv 2023.
[2] M. Oquab, et al. DINOv2: Learning Robust Visual Features without Supervision, arXiv 2023.
[3] A, Radford ,et al. Learning Transferable Visual Models From Natural Language Supervision, ICML 2021.

---

> ### Author Response · Authors · 2023-11-19
> **Part 1 of Response to Reviewer uqrh**
>
> **Dear Reviewer uqrh**
>
> We thank you for your comments and feedback. We hope to address your concerns here.
>
> **Weaknesses 1:** About Contribution and Novelty
>
> **A1:** Contribution and Novelty
>
> **Contribution:**
>
> Our core contribution is an exploration of how visual fundamental models can be applied to geometric vision tasks. We think this is illuminating and can be borrowed for other tasks, allowing non-semantic related tasks to benefit from the fundamental model wave as well.
>
> **Motivation and Novelty:**
>
> We systematically explore how the attractive properties of SAM can be utilized to guide local feature learning from three perspectives, intermediate representation, high-level semantic grouping, and low-level edge structure.
>
> 1. **Semantic relationship distillation (PSRD):**
>    Both segmentation (pixel-level classification) and local feature learning (pixel-level matching) can be viewed as pixel-level representation learning, and thus **the use of semantic discriminativeness to improve matching discriminativeness** is promising. However common semantic segmentation can only handle a limited number of categories, so we resort to the visual fundamental model SAM and category-independent semantic relationship distillation to break the category and scenario constraints.
>
> 2. **Semantic grouping (WSC):**
>    Instead of recognizing an entire object as a category (e.g., the pixels of the wheel and the window in Fig. 1 (a) are all identified as car), SAM can produce fine-grained part-level semantic groupings. These **semantic groupings can provide valuable weakly supervised signal supervision** (samples in same group are just weakly supervised positive samples, since they are not ground truth correspondences) for local feature learning, i.e. pixels belonging to the same semantic grouping should be closer in the description space, and on the contrary pixels of different groupings should be kept at a distance in the description space. In particular, to preserve local feature discriminability within the same semantic grouping, we add a fixed margin for positive sample optimization.
>
> 3. **Edge attention guidance (EAG):**
>    Edge regions are more worthy of the network’s attention than mundane regions. On one hand, corner and edge points in the edge region are more likely to be detected as keypoints. On the other hand, the edge region contains rich information about the geometric structure thus contributing more to the discriminative nature of the local descriptor. Therefore, we use the fine-grained boundaries distilled from the SAM as explicit attention map to direct the network to pay more attention to the boundary regions during local feature detection and description.
>
> Moreover, since the above improvements work only in the training phase, they do not introduce additional computational consumption in the inference phase.
>
> **Discussion:**
>
> We can understand your concerns; however, similarities in implementation details should not detract from the novelty of the methods themselves.
>
> 1. Our PSRD and [1] are just similar in implementation, but the motivation and physical meaning are different.
>    i) The goal of [1] is for teacher and student predictions to maintain consistent relationships between samples (mini batch-wise) whereas our PSDR maintains consistent relative relationships between pixel-level representations (pixel-wise).
>    ii) While most distillation methods focus on migrating knowledge from large teacher models to small models performing the same task, our PSRD introduces semantic discriminative properties of segmentation tasks (pixel-level classification) to pixel-level matching tasks.
>    iii) Similarly, CLIP [2] only utilized the common contrast learning loss implementation, but not to the detriment of his success and far-reaching impact.
>
> 2. WSC differs from comparative learning loss in that we set fixed boundaries between positive samples prompting it to optimize to a certain distance and no longer provide gradients. This has the advantage of ensuring pixel discriminability within the same semantic grouping.
>
> 3. To the best of our knowledge, explicit steering of networks via edge information has rarely been explored before.
>
> **Q1:** Intuition of each technique
>
> **A2:**
>
> 1. The motivation for each technique is described in answer 1 and in the "Introduction" section.
>
> 2. For the examples mentioned:
>    Unlike semantic segmentation, SAM does not project pixels to a specified semantic category, so the features extracted by the SAM encoder contain only category-independent semantic discriminative properties. This results in pixel features of the same class of objects (e.g., people) extracted from different images are not aligned. Direct distillation does not apply in this case (as reported in appendix section 1.4), so we resort to distilling the semantics contained in the encoder by exploiting the relative relationship between pixels (i.e., pixel representations of the same object are closer together).

---

> ### Author Response · Authors · 2023-11-19
> **Part 2 of Response to Reviewer uqrh**
>
> **Q2:** Is SAM really a good teacher?
>
> **A3:**
>
> 1. As stated in A1 and A2, SAM is category-independent segmentation and does not produce semantically aligned representations, which is consistent with the observation in Matcher [3]. However, we can solve this problem by performing category-independent semantic group distillation using semantic relationship.
>
> 2. SAM has part-level semantic grouping and boundary extraction capabilities not available in CLIP [2] and DINOv2 [4].
>
> 3. It is interesting to explore the distillation of other fundamental models, and we provide our insights in *appendix section 1.6*.
>
>
> **Q3:** A minor question of the use of SAM.
>
> **A4:**
>
> In our methodology, we leverage the automated mask generation function provided by SAM (Segment Anything Model) An example can be found: [https://github.com/facebookresearch/segment-anything/blob/main/notebooks/automatic_mask_generator_example.ipynb](https://github.com/facebookresearch/segment-anything/blob/main/notebooks/automatic_mask_generator_example.ipynb). Using the default automated mask generation function, SAM automatically generates an isometric grid on the image. Each point is used as a hint, and SAM can predict multiple masks from each hint. Then, non-maximal suppression was used to filter and optimize the mask results. In that case, SAM, by default, tries to segment everything in the input image. Default hyper-parameters are used. Details could be referred to [https://github.com/facebookresearch/segment-anything/blob/main/segment_anything/automatic_mask_generator.py](https://github.com/facebookresearch/segment-anything/blob/main/segment_anything/automatic_mask_generator.py) This methodology ensures the extraction of high-quality and distinct semantic groupings, forming a foundational element in our approach to local feature learning.
>
> **Q4:** Can we directly use SAM feature?
>
> **A5:**
>
> Using SAM features directly for image matching and visual localization tasks raises specific challenges. The segmentation function of SAM aims to assign semantic categories to each pixel, which may not align perfectly with the pixel-level correspondences required for matching tasks. This mismatch can lead to intra-class matching errors, where pixels belonging to the same semantic class may be erroneously assigned similar representations.
>
> In our experimentation on Hpatches, direct matching using SAM features yielded a performance of only 0.32 MMA@3, highlighting the limitations of utilizing SAM features alone for these tasks.
>
> Additionally, it's important to note that SAMFeat is designed for efficiency. SAM is only used in the training stage via Knowledge distillation. This optimized SAMFeat's inference speed for efficient deployment in real-world scenarios. Directly using the SAM feature in the image matching or visual localization inference stage will lead to low efficiency and inference speed because SAM's encoder backbone is heavily parameterized compared to our backbone.
>
> **Q5:** First line of table 3.
>
> **A6:**
>
> Our sincere apologies for not elaborating clearly and causing confusion to you, and we have corrected this in the main paper. In short, our baseline, the first line of the main paper table 3, is not the exact MTLDesc. To enhance clarity and simplicity, our baseline adopts SuperPoint's VGG-styled backbone with MTLDesc's attention-weighted descriptor loss. MTLDesc used complex modules such as Transformer to achieve high matching accuracy. Thus, with no doubt, our baseline (75.7 MMA@3 in the main paper table 3) is lower in performance compared to MTLDesc (78.7 MMA@3 in the main paper table 1). Compared to the MTLDesc which has 10.1 Million trainable parameters, SAMFeat is much more lightweight and only has 2.1 Million trainable parameters.
>
> [1] F, Tung, et al. Similarity-Preserving Knowledge Distillation, ICCV19
>
> [2] A, Radford ,et al. Learning Transferable Visual Models From Natural Language Supervision, ICML 2021.
>
> [3] Y,Liu, et al. Matcher: Segment Anything with One Shot Using All-Purpose Feature Matching, arXiv 2023.
>
> [4] M. Oquab, et al. DINOv2: Learning Robust Visual Features without Supervision, arXiv 2023.

---

### Official Review · Reviewer_F7AB · 2023-10-31

**Soundness:** 3 good
**Presentation:** 3 good
**Contribution:** 3 good
**Rating:** 6
**Confidence:** 4

**Summary:**

In this paper, the authors introduce SAMFeat, leveraging SAM (segment anything model) as a teacher to enhance local feature learning to promote the performance of local feature detection and description. In practice, SAMFeat uses Pixel Semantic Relational Distillation (PSRD) to distill feature relations with category-agnostic semantic information learned by the SAM for improved feature description. In addition, a technique called Weakly Supervised Contrastive Learning Based on Semantic Grouping (WSC) is adopted for optimizing the metric space of local descriptors. At last, an Edge Attention Guidance Module (EAGM) enhances detection accuracy by focusing on edge regions guided by SAM. The experimental results demonstrate the effectiveness of the proposed modules.

**Strengths:**

(1) The authors integrate the strengths of existing frameworks, effectively utilizing the SAM foundation model and successfully distilling its knowledge into the network for local descriptor learning. It is a good paper for leveraging the knowledge of large models to enhance domain-specific tasks effectively.

(2)The article is clearly written in most parts, enabling readers to quickly catch up on the core technical points. The proposed approach is quite reasonable.

(3)The experimental results demonstrate significant technical advantages, showing substantial improvements over previous work.

**Weaknesses:**

(1) My first concern is about the novelty of the paper. It is commendable to leverage SAM to enhance model performance in corresponding tasks. However, acquiring structured information through SAM (PSRD), and using semantic grouping to construct positive and negative samples, thereby introducing contrastive learning, have already been briefly discussed in previous works (SFD2, TPR). From this, the paper is more like an integration of some schemes combined with the SAM  model. Hence, its technical novelty is somewhat weak from my perspective.

(2) In practical applications, is it really necessary to adopt all the components described in the paper? The actual contributions of different losses to model optimization still need further verification. For example, in Table 3, how about the performance without using EAG? In addition, are WCS and EAG really irreplaceable by each other? For instance, could eliminating WCS and adjusting the weight of EAG not affect the final results? This raises concerns about the actual importance of WCS ( the semantic embedding ) in the current framework.

(3) Again, regarding Table 4 in the Appendix, the performance seems quite sensitive to the choice of hyperparameters. Would adjusting the loss weight of EAG also further affect the choice of these hyperparameters? After all, as mentioned in the main article, the features enhanced by EAG are further used for subsequent WCS calculations.

(4) Some technical implementation details are still unclear. For example, for Local Feature Detection, is the enhanced C3 computed first before proceeding to calculate C4? The structural diagram in Figure 2 and the description in the subsection on Local Feature Detection have caused confusion regarding more specific technical details. I hope the authors can polish this part.

**Questions:**

(1) See the weakness

(2) As mentioned in the introduction, how to obtain a large number of effective annotated samples for local descriptor learning is still very challenging. Could it be possible to demonstrate through experiments that the proposed method can effectively reduce the requirement of the training samples?

**Details Of Ethics Concerns:**

From my viewpoint, there are no ethical concerns.

---

> ### Author Response · Authors · 2023-11-19
> **Response to Reviewer F7AB**
>
> Dear Reviewer F7AB,
>
> Thank you for your valuable feedback. We hope to address your concerns as follows:
>
> **Weaknesses1:**
> Novelty of SAMFeat
>
> **A1:**
> Thank you for the reminder. We have further emphasized the differences from existing approaches in related work.
>
> **Differences from existing methods:**
>
> **1) Motivation.** Although some schemes (SFD2, SeLF) have used semantic segmentation information to improve the performance of local features on visual localization tasks, they perform poorly on tasks other than visual localization, as shown in the main paper Table 1 and the "Limitations" section reported in the SFD2. Semantic segmentation can only segment certain **specific categories** (e.g., visual localization-related street scenes), preventing such approaches from generalizing to other scenarios. In contrast, our approach aims to make semantic information accessible to any scenario by subtly leveraging SAM **category-independent** semantic information (representation relations, semantic grouping at the part level, boundaries).
>
> **2) Implementation.** Our approach differs significantly from existing methods in its concrete implementation. **i) semantic distillation:** Unlike semantic segmentation, SAM does not project pixels to a specified semantic category, so the features extracted by the SAM encoder contain only category-independent semantic discriminative properties. This results in pixel features of the same class of objects (e.g., people) extracted from different images are not aligned. Direct distillation (like SFD2, SeLF) does not apply in this case (as reported in appendix section 1.4), so we resort to distilling the semantics contained in the encoder by exploiting the relative relationship between pixels (i.e., pixel representations of the same object are closer together). **ii) semantic grouping:** SFD2 constructs the metric learning loss according to each specific category (e.g., cars, people). It has two problems, on the one hand, it can only work on a limited number of categories, and on the other hand, category-level semantic grouping is not reliable for local feature learning. For example, the pixels of wheels and windows in Fig. 1 (a) are recognized as cars, which is not conducive to discriminative learning of local features. In contrast, our proposed WSC is category-independent and uses part-level semantic grouping. **iii) edges:** To the best of our knowledge, existing schemes rarely explore the improvement of boundary information for local feature learning.
>
> **W2:**
> Necessity of each component
>
> **A2:**
> Thanks to your suggestion, we performed a more thorough ablation study in Appendix Table 7. Specifically, we report the results of removing each component individually in addition to applying the components sequentially.  Since the three components exert influence from different perspectives (intermediate representations, high-level semantic segmentation, and low-level edge structure), working together produces optimal performance.
>
> **W3:**
> Ablation in loss weights
>
> **A3:**
> Thanks to your suggestion, we have added an exploration of EAG loss weights in the Appendix Table 8. There are only subtle changes of about 0.2 on the MMA@3 as we tune the EAG loss weights. The minor changes in the MMA@3 show that it is difficult to achieve the effect of imposing WCS by only adjusting the loss weights. It is not so difficult to allow the network to extract boundaries, as shown in Figure 1 of the appendix. Therefore the effect of adjusting the EAG loss weights on the final performance is slight.
>
> **W4:**
> More technical details
>
> **A4:**
> Thank you for the reminder, and we provided a more precise description of this process. We enhance $C_{3}$ only when the $\rm concat$ feature maps are fed into the detection head (because only the shape of $C_{3}$ is consistent with the edge map), so $C_{4}$ is not related to the enhanced $C_{3}$. In addition, we will release the code that precisely specifies the implementation details to facilitate further understanding by the reader.
>
> **Q2:**
> Robustness to training sample size
>
> **A5:**
> Thanks for your suggestion. **(1)** We add a comparison of training data sizes with advanced methods in the Appendix Table 5. The results show that our method uses less data and is uncorrelated with the test benchmarks, achieving the best performance due to the introduction of the additional supervision provided by SAM. **(2)** Due to time constraints, we will leave more exploration of SAMFeat's role in mitigating data dependencies for future work. Since it is cheap for SAM to provide supervision, we will collect a large amount of web data and mix it with the original dataset to train SAMFeat V2. We believe that the SAMFeat V2 enhancement should be more in terms of generalization; however, the current benchmark lacks the relevant evaluation. We will collect more image pairs of different types and build a large open-world image-matching benchmark to evaluate current methods.

---

### Official Review · Reviewer_SX6K · 2023-11-02

**Soundness:** 2 fair
**Presentation:** 2 fair
**Contribution:** 2 fair
**Rating:** 5
**Confidence:** 3

**Summary:**

In this paper, the authors proposed to leverage SAM as the teacher model for learning local feature detectors and descriptors. The authors argued that the fine-grained masks generated by SAM can provide a good amount of prior information about the image and thus be beneficial to local feature learning. To guide the local feature learning, the authors proposed to extract the information from three aspects: 1) pixel-wise representation relationship; 2) semantic group and 3) edge map from the SAM outputs. Based on this information, the authors developed a distillation objective function to bridge the gap between SAM outputs and the local feature detector. In the experiments, the authors showed that the proposed method of distilling from SAM can improve the performance on HPatches and Aachen V1.1. The ablation also justified the effectiveness of each component in the proposed model.

**Strengths:**

1. The authors proposed to leverage the foundational segmentation model SAM for local feature learning. As highlighted in the paper, this work is the first one that incorporates SAM for local feature learning by distilling the knowledge from SAM.

2. The authors proposed three techniques to transfer the fine-grained image understanding knowledge from SAM to the proposed local feature learning pipeline, which results in a new local feature detector called SAMFeat.

3. The experimental results on two benchmarks HPatches and Aachen demonstrate the effectiveness of the proposed method for local feature detection and matching.

**Weaknesses:**

1. The proposed method in this work is heuristic and incremental. Though the combination of all three techniques achieves the best performance, it is not clear how each of the heuristic technique improve the local feature learning and further the final performance. I would highly suggest the authors have a deeper study on the proposed techniques on how they are contributing the final performance.

2. It is not clear how much overhead for the training after adding the extra loss functions. For example, if I understand correctly, the relationship distillation seems to be very computational heavy in that the affinity matrix contains quadratic number of entries to the image size. Likewise, the second loss term need to compute the with-group and cross-group similarities for all image regions. Overall, I am afraid that the proposed method is very time consuming during training.

3. The experiments in the paper is not satisfactory. There is not much detailed and deep analysis on the proposed method as I mentioned earlier. Given the two tables Table 1 and 2, it is hard to tell whether the proposed method is fairly compared with previous works. For example, the authors mentioned that they used MTLDesc as the baseline method. In Table 1, the reported MMA@3 is 78.7 for it. However, when it goes to Table 3, the first row shows 75.7 MMA@3. This makes me a bit confused and doubted whether the authors are conducting solid and fair comparisons across the board. More importantly, it is also not clear about the main difference between the proposed method and others, regarding the learning techniques, vision encoder, training regimes, etc.

4. Following the last point, the incorporation of SAM for local feature learning is interesting and valuable. However, it is not clear how the settings for the distillation affect the final performance. For example, the density of grid in SAM for automatical segmentation, the image resolutions, the number of sampling points for the proposed training losses, etc, all of them are not studied, which make the contribution of the work and effectiveness of the proposed method hard to assess.

**Questions:**

Like I mentioned above, I do want to see a deeper analysis on the proposed techniques to distill SAM knowledge for local feature learning tasks.

---

> ### Author Response · Authors · 2023-11-19
> **Part 1 of Response to Reviewer SX6K**
>
> Dear Reviewer SX6K,
>
> We thank you for your valuable feedback. We would like to respond to your comments to address your concerns here.
>
> **Weakness 1:** Deeper study on the proposed techniques
>
> **A1:**
>
> 1. **How each component works:**
>    We systematically explore how the attractive properties of SAM can be utilized to guide local feature learning from three perspectives, intermediate representation, high-level semantic grouping, and low-level edge structure. We have designed three interesting techniques to bootstrap local feature learning by taking advantage of the three attractive properties of SAM.
>
>    1. **Semantic relationship distillation (PSRD):**
>       Both segmentation (pixel-level classification) and local feature learning (pixel-level matching) can be viewed as pixel-level representation learning, and thus **the use of semantic discriminativeness to improve matching discriminativeness** is promising. However, common semantic segmentation can only handle a limited number of categories, so we resort to the visual fundamental model SAM and category-independent semantic relationship distillation to break the category and scenario constraints.
>
>    2. **Semantic grouping (WSC):**
>       Instead of recognizing an entire object as a category (e.g., the pixels of the wheel and the window in main paper Fig. 1 (a) are all identified as car), SAM can produce fine-grained part-level semantic groupings. These **semantic groupings can provide valuable weakly supervised signal supervision** (samples in the same group are just weakly supervised positive samples since they are not ground truth correspondences) for local feature learning, i.e. pixels belonging to the same semantic grouping should be closer in the description space, and on the contrary, pixels of different groupings should be kept at a distance in the description space. In particular, to preserve local feature discriminability within the same semantic grouping, we add a fixed margin for positive sample optimization.
>
>    3. **Edge attention guidance (EAG):**
>       Edge regions are more worthy of the network’s attention than mundane regions. On one hand, corner and edge points in the edge region are more likely to be detected as keypoints. On the other hand, the edge region contains rich information about the geometric structure thus contributing more to the discriminative nature of the local descriptor. Therefore, we use the fine-grained boundaries distilled from the SAM as an explicit attention map to direct the network to pay more attention to the boundary regions during local feature detection and description.
>
>    Moreover since the above improvements work only in the training phase, they do not introduce additional computational consumption in the inference phase.
>
> 2. **Validity Exploration:**
>    We performed a thorough ablation study to verify the effect of each component on SAMFeat in appendix Table 7. Specifically, we report the results of removing each component individually in addition to applying the components sequentially. Experimental results show that each of our proposed components can independently improve the performance of the method.
>
> **Weakness 2:** Not clear about training cost
>
> **A2:**
>
> 1. Even though the computation cost for the relationship matrix distillation is quadratic, we downsampled the feature map to 1/8 of the original size. This also quadratically reduced the computation cost.
>
> 2. In terms of quantitative analysis, our method only requires training for 6 hours using two Nvidia RTX 3090 GPUs. Compared to other work like ASLFeat (42 hours on a single NVIDIA RTX 2080Ti) and TRR (30 hours for training with two NVIDIA-A100 GPUs), this demonstrates a totally reproducible cost for individual researchers. As for the overhead for the training after adding the extra loss functions, we provide a detailed table in Appendix Table 6. Each loss function will inevitably cause extra training costs; however, the tradeoff between a minimal incremental in time and the improvement in accuracy is reasonable, and the final training cost is acceptable. This further demonstrates the lightweight nature of our approach in the field of feature learning and description, making it easily implementable and resource-efficient.

---

> ### Author Response · Authors · 2023-11-19
> **Part 2 of Response to Reviewer SX6K**
>
> **Weakness 3:** About experiments
>
> **A3:**
>
> Our experiments are conducted fairly.
>
> 1. For the confusion about Table 1 and Table 3, we sincerely apologize for not elaborating clearly and causing confusion to you, and we have corrected this in the main paper. In short, our baseline, the first line of the main paper table 3, is not the exact MTLDesc. To enhance clarity and simplicity, our baseline adopts SuperPoint's VGG-styled backbone with MTLDesc's attention-weighted descriptor loss. MTLDesc used complex modules such as Transformer to achieve high matching accuracy. Thus, with no doubt, our baseline (75.7 MMA@3 in main paper table 3) is lower in performance compared to MTLDesc (78.7 MMA@3 in main paper table 1) Compared to the MTLDesc which has 10.1 Million trainable parameters, SAMFeat is much more lightweight and only has 2.1 Million trainable parameters.
>
> 2. Regarding the fairness of our experiment, we ensure fairness in our evaluations; each method undergoes the same assessment criteria in HPatches and Aachen V1.1. Our Evaluation code is attached in the supplementary materials and is open-sourced. We ensure full transparency and fairness in evaluation and comparison. While each method may have unique training settings, the parameters of the visual encoders and the sizes of the training datasets are comparable. From the perspective of the visual encoder, compared to the MTLDesc which has 10.1 Million trainable parameters, SAMFeat is much more lightweight and only has 2.1 Million trainable parameters. Regarding the training dataset aspect, please refer to Appendix Table 5 for a comprehensive report on our dataset specifics. Our SAMFeat used fewer training samples and is more lightweight, but achieves the best performance. It's noteworthy that our training methods align with SuperPoint and MTLDesc, providing consistency and comparability in the training process.
>
> **Weakness 4 and Q1:** Deep analysis on SAM distillation.
>
> **A4:**
>
> Thank you for your suggestion. We acknowledge the importance of studying the impact of various settings on distillation. In our experiments, we adhered to the default settings of SAM, including the grid density. The training settings, such as image resolution and the number of sampling points, were kept consistent with MTLDesc to ensure a fair and direct comparison since the training data were identical. Notably, using these uniform and default settings resulted in a significant performance boost. Thus, the results obtained without meticulously fine-tuning each parameter underscore the effectiveness of our approach. In the appendix, we provide a detailed ablation study (Appendix Table 2 - Table 8) regarding the effect of different distillation loss settings and the effect of each distillation and learning module on the final outcome. We hope this could address your concerns. As you rightly pointed out, due to time constraints, we couldn't explore every parameter in-depth, and in many cases, we followed established practices from previous works.

---

### Author Response · Authors · 2023-11-20
**General Response**

**General Response:**

We express our sincere gratitude to each reviewer for their thoughtful feedback on our paper, "Segment Anything Model is a Good Teacher for Local Feature Learning." We appreciate the time and effort dedicated to evaluating our work and have thoroughly considered all comments. In addition to individual responses, this comprehensive reply summarizes our main contributions, motivation, strengths acknowledged by reviewers, and changes made in our updated SAMFeat paper and appendix according to the reviewers’ feedback.

**1. Motivation and Contribution:**

Our primary contribution is an exploration of applying visual fundamental models to geometric vision tasks, illuminating potential applications for non-semantic-related tasks. We systematically leverage SAM's attractive properties to guide local feature learning across intermediate representation, high-level semantic grouping, and low-level edge structure. Introducing SAMFeat, we utilize SAM as a teacher to enhance local feature learning, employing Pixel Semantic Relational Distillation (PSRD), Weakly Supervised Contrastive Learning Based on Semantic Grouping (WSC), and the Edge Attention Guidance Module (EAGM). Experimental results affirm the effectiveness of our proposed modules.

**2. Strengths Acknowledged by Reviewers:**

We appreciate the positive recognition of our work:
1. **Innovative Use of SAM:** Pioneering the integration of SAM for local feature learning marks a significant contribution (reviewers SX6K, F7AB, uqrh, and 933k).
2. **Effectiveness Demonstrated Through Experiments:** Strong evidence on HPatches and Aachen benchmarks supports the method's effectiveness in local feature detection and matching, contributing to state-of-the-art advancements (Reviewers SX6K, F7AB, uqrh).
3. **Open-Source Code Availability:** Providing open-source code enhances reproducibility and accessibility, contributing to the paper's strength (Reviewer uqrh).
4. **Clarity in Presentation:** The article's clear and comprehensible writing facilitates understanding, contributing to its overall strength in presentation (Reviewers F7AB, 933k).

**3. Improvements in the Updated Version:**

In response to reviewers' feedback, we have made several improvements in the updated version of SAMFeat and the Appendix:
1. Clarification and polish of technical and experimental details.
2. Addition of ablation experiments, training data comparisons, and detailed elaboration on SAMFeat technical modules in the Appendix.
3. Inclusion of insights and analysis on the distillation of other fundamental models in the Appendix.

**4. Individual Responses:**

For more specific feedback, we will address each concern in response to the respective reviewer's comments.

We value the constructive feedback, which will significantly contribute to refining our paper. We are dedicated to addressing concerns and enhancing the clarity, rigor, and novelty of our work.

Thank you!

Best regards,

Authors from SAMFeat

---

### Meta-Review · Area_Chair_4aYG · 2023-12-05

**Metareview:**

This paper presents an approach utilizing the features extracted by SAM to guide the learning of local features.

After the rebuttal and AC-reviewer discussion stage, the final scores of this paper are 5/5/5/6. In the discussion stage, three reviewers offered their feedback. Two negative reviewers (rating 5) insisted on rejection, and one positive reviewer (rating 6) said that the paper lacks novelty and is incremental -- he gave a score of 6 but he does not work on this area and he did not mind rejection. The AC also took the authors' private comments into consideration but found no reason to overturn the overall negative recommendation.

**Justification For Why Not Higher Score:**

The reviewers arrived at a consensus of rejection.

**Justification For Why Not Lower Score:**

N/A

---

### Decision · Program_Chairs · 2024-01-16

Reject